# Preventive effects of nitrate-rich beetroot juice supplementation on monocrotaline-induced pulmonary hypertension in rats

**Masashi Tawa**[1,2]*, **Rikako Nagata**[1◉], **Yuiko Sumi**[1◉], **Keisuke Nakagawa**[1], **Tatsuya Sawano**[1,3], **Mamoru Ohkita**[1], **Yasuo Matsumura**[1]

**1** Laboratory of Pathological and Molecular Pharmacology, Osaka University of Pharmaceutical Sciences, Takatsuki, Osaka, Japan, **2** Department of Pharmacology, Kanazawa Medical University, Kahoku, Ishikawa, Japan, **3** Division of Molecular Pharmacology, Faculty of Medicine, Tottori University, Yonago, Tottori, Japan

◉ These authors contributed equally to this work.
* masashi.tawa@ompu.ac.jp, tawa@kanazawa-med.ac.jp

**Data Availability Statement:** All relevant data are within the manuscript and its Supporting information files.

## Abstract

Beetroot (*Beta vulgaris L.*) has a high level of nitrate; therefore, its dietary intake could increase nitric oxide (NO) level in the body, possibly preventing the development of pulmonary hypertension (PH). In this study, we examined the effects of beetroot juice (BJ) supplementation on PH and the contribution of nitrate to such effects using a rat model of monocrotaline (MCT, 60 mg/kg s.c.)-induced PH. Rats were injected subcutaneously with saline or 60 mg/kg MCT and were sacrificed 28 days after the injection. In some rats injected with MCT, BJ was supplemented from the day of MCT injection to the day of sacrifice. First, MCT-induced right ventricular systolic pressure elevation, pulmonary arterial medial thickening and muscularization, and right ventricular hypertrophy were suppressed by supplementation with low-dose BJ (nitrate: 1.3 mmol/L) but not high-dose BJ (nitrate: 4.3 mmol/L). Of the plasma nitrite, nitrate, and their sum (NOx) levels, only the nitrate levels were found to be increased by the high-dose BJ supplementation. Second, in order to clarify the possible involvement of nitrate in the preventive effects of BJ on PH symptoms, the effects of nitrate-rich BJ (nitrate: 0.9 mmol/L) supplementation were compared with those of the nitrate-depleted BJ. While the former exerted preventive effects on PH symptoms, such effects were not observed in rats supplemented with nitrate-depleted BJ. Neither supplementation with nitrate-rich nor nitrate-depleted BJ affected plasma nitrite, nitrate, and NOx levels. These findings suggest that a suitable amount of BJ ingestion, which does not affect systemic NO levels, can prevent the development of PH in a nitrate-dependent manner. Therefore, BJ could be highly useful as a therapy in patients with PH.

## Introduction

Nitric oxide (NO) is a biological molecule that is derived either through the L-arginine-NO synthase-NO pathway or nitrate-nitrite-NO pathway [1–3]. Its proven vasodilatory, antithrombotic, anti-fibrotic, and anti-proliferative effects [1–3] explain why a decrease in NO

**Funding:** The authors received no specific funding for this work.

**Competing interests:** The authors have declared that no competing interests exist.

bioavailability is closely associated with the development of various diseases such as pulmonary hypertension (PH) that is characterized by obstructive lesions in small pulmonary arteries, increased pulmonary arterial pressure, and right ventricular hypertrophy [4]. Due to the poor prognosis of this condition, it strongly warrants an early diagnosis with an effective therapeutic intervention. Drugs that target the NO signaling pathway are currently recommended as therapy in patients with PH to compensate for the decreased NO levels in pulmonary circulation [5–7].

Beetroot (*Beta vulgaris L.*) is a vegetable rich in nitrates, polyphenols, flavonoids, organic acids, and so on [8]. Studies have shown beetroot to elicit similar effects as inorganic nitrate when ingested [9–11]. In this context, beetroot juice (BJ) intake has been revealed to lower blood pressure by increasing the level of NO through the nitrate-nitrite-NO pathway [12–14]. In addition, dietary BJ ingestion has been reported to reduce arterial wave reflections, which, in a nitrate-dependent manner, is associated with left ventricular remodeling in chronic heart failure [15]. These findings suggest the possible benefits of BJ supplementation even in PH. In fact, recent animal and human studies have both shown BJ supplementation to be an useful option in preventing PH [16, 17]. Importantly, however, although treatment with inorganic nitrate has been demonstrated to improve PH symptoms [18–20], it is still unclear whether the preventive effects of BJ supplementation on PH are dependent on nitrate. Therefore, the aim of this study was to address this issue by investigating the effects of nitrate-rich and nitrate-depleted BJ in a rat model of monocrotaline (MCT)-induced PH.

## Materials and methods

### Animals

The Experimental Animal Research Committee of Osaka University of Pharmaceutical Sciences provided ethical approval for the laboratory animals used in this study (Permit No: 6/2018). A total of 57 male Sprague-Dawley rats (8-week-old) were obtained from Japan SLC, Inc. (Shizuoka, Japan). The rats were allowed free access to food and water in a light-controlled room through a 12-h light-dark cycle at the Animal Testing Facility, Osaka University of Pharmaceutical Sciences.

### Experimental protocols

In the first experiment, the dose-related effects of nitrate-rich BJ (Beet It Organic, James White Drinks Ltd., Ipswich, United Kingdom) supplementation were examined to determine the appropriate dose of BJ. Briefly, rats were administered a subcutaneous injection of physiological saline (referred to as "sham") or 60 mg/kg MCT (Sigma Chemical Co., St. Louis, MO). While saline-injected rats received water, MCT-injected rats were supplemented, from the day of MCT injection, with either water, low dose BJ (containing 1.3 mmol/L nitrate with a dilution factor of 30), or high dose BJ (containing 4.3 mmol/L nitrate with a dilution factor of 10) and were tagged "MCT", "LD BJ", and "HD BJ", respectively. Water was used as the diluent for the two different BJ doses; these doses were determined following our previous study [16]. The solutions were changed every 2–3 days. The average water intake per day was as follows: sham group, 37 mL; MCT group, 35 mL; LD BJ group, 31 mL (nitrate intake: 133 µmol/kg/day); and HD BJ group, 32 mL (nitrate intake: 451 µmol/kg/day).

In the second experiment, the respective effects of nitrate-rich BJ (Beet It Sports Shots, James White Drinks Ltd.) and nitrate-depleted BJ (Placebo Shots, James White Drinks Ltd.) supplementation were examined to establish the contribution of nitrate to the preventive effects of BJ. From the first day of MCT injection, the rats were administered nitrate-rich BJ (containing 0.9 mmol/L nitrate with a dilution factor of 100) or nitrate-depleted BJ (with a

dilution factor of 100) and were tagged "BJ" and "PJ", respectively. With reference to the results obtained from the first experiment, the dilution ratio of the nitrate-rich BJ was calculated based on the nitrate concentration. Other procedures were exactly repeated as described above. The average water intake per day was as follows: sham group, 42 mL; MCT group, 38 mL; BJ group, 39 mL (nitrate intake: 117 μmol/kg/day); and PJ group, 45 mL.

At 28 days after the initiation of treatment, hemodynamic parameters of the rats that survived to that point were measured, and the blood samples were collected from the inferior vena cava to determine the plasma nitrite, nitrate, and their sum (NOx) concentrations. The rats were then euthanized, and the heart and lungs were excised, weighed, and finally subjected to a morphometric analysis. It is worthy of mention that the rats were not deprived food before the experiments.

### Hemodynamic measurement

The rats were anesthetized by an intraperitoneal injection of sodium pentobarbital (40 mg/kg; Kyoritsu Seiyaku Co., Tokyo, Japan). After a state of stable anesthesia was maintained, the right carotid artery was isolated, incised, and a polyethylene catheter connected to a pressure transducer was inserted to measure central hemodynamics; heart rate and mean arterial pressure were both monitored using a polygraph system (RM 6000, Nihon Kohden, Tokyo, Japan). Another polyethylene catheter was inserted into the right ventricle via the right jugular vein to record the right ventricular systolic pressure (RVSP).

### Histological analysis

The left lung was inflated by injection of 10% phosphate-buffered formalin, fixed in the formalin, embedded in paraffin, and sectioned at a thickness of 4 μm. The sections were stained with Elastica van Gieson and examined under a light microscope. Vessels that have two well-defined elastic lamellae with a layer of smooth muscle in between were defined as resistance pulmonary arteries. The external diameter and medial wall thickness were determined for 10–15 muscular arteries (50–100 μm in external diameter) per lung section using a computer-assisted image analyzer (AE-6905C, ATTO, Tokyo, Japan). The percent wall thickness was calculated for each artery and expressed as follows: medial thickness (%) = [(medial thickness × 2)/ external diameter] × 100. For each animal, the average of the values obtained was estimated and used for the calculation of group means.

The degree of microvascular muscularization in more than 30 vessels (of external diameter < 50 μm) per lung section stained with Elastica van Gieson was determined, according to a previously reported method [21]. The vessels were categorized as non-muscularized (< 10% medial coat of muscle) or muscularized (≥ 10% medial coat of muscle), and the latter was defined as remodeled vessels. The proportion of the remodeled vessels to total vessels was determined for each animal and used to calculate the group means.

### NOx measurement

Plasma was deproteinized by mixing with an equal volume of methanol. The resulting mixture was then centrifuged at $10,000 \times g$ for 10 minutes at 4˚C; the supernatant obtained was stored at −80˚C for later analysis. NOx concentration was measured using a high-performance liquid chromatography-Griess system (ENO-20, Eicom, Kyoto, Japan). Briefly, the supernatant (10 μL) was diluted with the mobile phase (NOCARA, Eicom) and injected into the system, wherein the analysis was conducted by measuring the absorbance of azo dye formed through a "diazotization-coupling" reaction. Nitrite and nitrate levels were determined through comparison with the results obtained using standard solutions (NO-STD, Eicom).

## Statistics

All data are presented as individual values and/or means ± SEM. Statistical analyses were performed using the Graph Pad Prism software (version 7.0, Graph Pad Software Inc., San Diego, CA). Data were finally compared using the Sidak multiple comparisons tests following the one-way analysis of variance. A value of $p < 0.05$ was considered significant.

## Results

### Effects of nitrate-rich BJ supplementation on PH symptoms

MCT injection led to an elevation in RVSP, which was observed to be significantly suppressed in the LD BJ group (Fig 1a). In addition, pulmonary arterial medial thickening and muscularization (Fig 1b) and right ventricular hypertrophy (Fig 1c), induced by MCT injection, were observed to be mild in the LD BJ group. On the contrary, such beneficial effects on the MCT-induced PH symptoms were not observed in the HD BJ group (Fig 1a–1c). Central hemodynamics was not significantly different among the sham, MCT, LD BJ, and HD BJ groups (Table 1).

### Effects of nitrate-rich BJ supplementation on NO levels

There were no significant differences in plasma nitrite (Fig 2a), nitrate (Fig 2b), and NOx concentrations (Fig 2c) between the sham and MCT groups. While the nitrate concentration in the HD BJ group was significantly higher than that in the sham group; other concentrations of

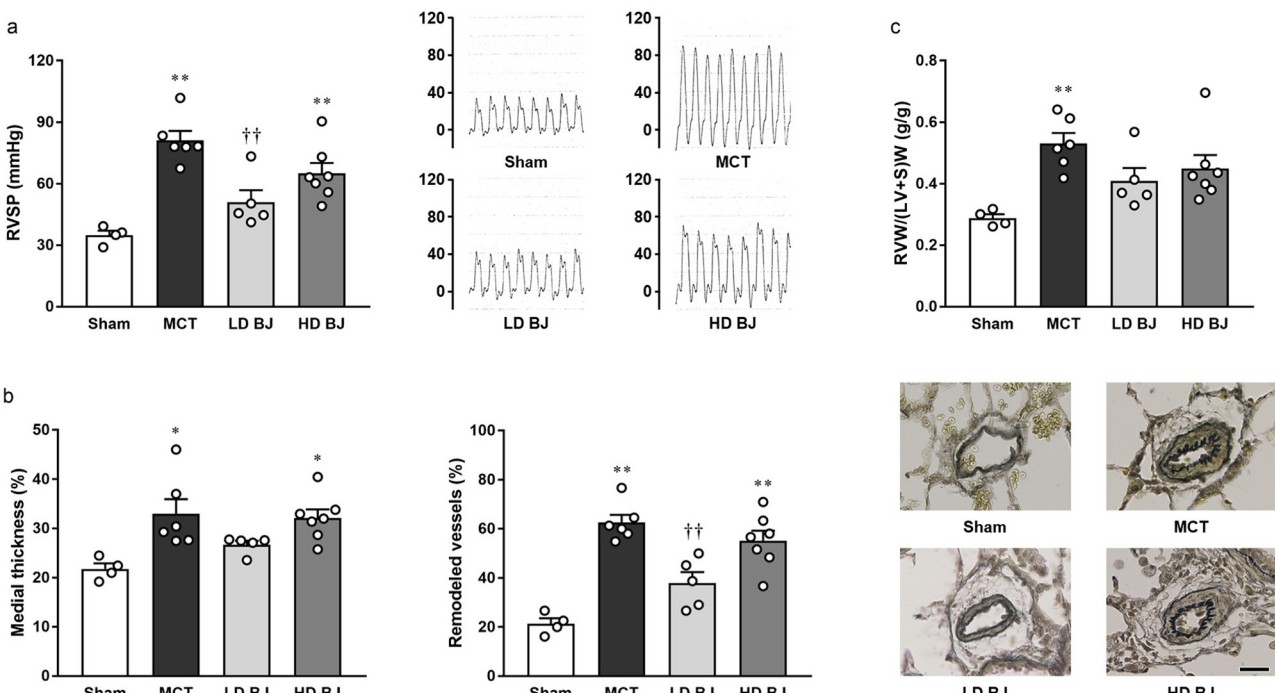

**Fig 1. Effects of nitrate-rich BJ supplementation on PH symptoms in MCT-injected rats.** (a) RVSP (left panel) and typical tracings (right panel). (b) Pulmonary arterial medial thickening (left panel), remodeling (middle panel), and typical images (right panel; scale bar, 25 μm). (c) RVW/(LV+S)W. Each point represents individual value; each column and bar represent the mean ± SEM values of 4–7 experiments. *p < 0.05 and **p < 0.01, compared to the sham group; ††p < 0.01, compared to the MCT group. Statistical analysis was performed using one-way analysis of variance with Sidak post hoc test. Abbreviations: BJ, beetroot juice; HD, high-dose; LD, low-dose; MCT, monocrotaline; RVSP, right ventricular systolic pressure; RVW/(LV+S)W, right ventricle-to-left ventricle plus septum weight ratio.

**Table 1. Effects of nitrate-rich BJ supplementation on various parameters.**

|  | Sham (n = 4) | MCT (n = 6) | LD BJ (n = 5) | HD BJ (n = 7) |
|---|---|---|---|---|
| BW (g) | 362 ± 8 | 330 ± 8 | 358 ± 13 | 350 ± 7 |
| HW/BW (g/kg) | 2.55 ± 0.02 | 3.20 ± 0.10** | 2.82 ± 0.13 | 2.81 ± 0.08† |
| RVW/BW (g/kg) | 0.52 ± 0.02 | 1.03 ± 0.06** | 0.77 ± 0.09 | 0.80 ± 0.06 |
| (LV+S)W/BW (g/kg) | 1.82 ± 0.02 | 1.94 ± 0.04 | 1.87 ± 0.04 | 1.79 ± 0.05 |
| LW/BW (g/kg) | 3.40 ± 0.08 | 6.27 ± 0.37** | 5.22 ± 0.36** | 5.25 ± 0.24** |
| HR (bpm) | 332 ± 49 | 409 ± 27 | 335 ± 41 | 369 ± 13 |
| MAP (mmHg) | 88 ± 6 | 88 ± 7 | 90 ± 6 | 90 ± 6 |

Data are the mean ± SEM values of 4–7 experiments.

**p < 0.01, compared to the sham group;

†p < 0.05, compared to the MCT group.

Statistical analysis was performed using one-way analysis of variance with Sidak post hoc test. Abbreviations: MCT, monocrotaline; BJ, beetroot juice; LD, low-dose; HD, high-dose; BW, body weight; HW, heart weight; RVW, right ventricular weight; (LV+S)W, left ventricular plus septum weight; LW, lung weight; HR, heart rate; MAP, mean arterial pressure.

NO metabolites in the LD BJ and HD BJ groups did not differ from those in the sham and MCT groups (Fig 2a–2c).

### Effects of nitrate-depleted BJ supplementation on PH symptoms

RVSP in the BJ group was significantly lower than that in the MCT group; this effect, however, was neutralized by nitrate depletion as shown in the PJ group (Fig 3a). In addition, unlike the BJ group, where no significant pulmonary arterial medial thickening and muscularization were detected, apparent thickening and muscularization were observed in the PJ group (Fig 3b). Additionally, unlike the BJ group, marked right ventricular hypertrophy, similar to that observed in the MCT group, was noted in the PJ group (Fig 3c). It is also worthy of notable mention that central hemodynamics was uniform across all groups (Table 2).

### Effects of nitrate-depleted BJ supplementation on NO levels

Plasma concentrations of NO metabolites, including nitrite (Fig 4a), nitrate (Fig 4b), and NOx (Fig 4c), were not significantly different among the sham, MCT, BJ, and PJ groups.

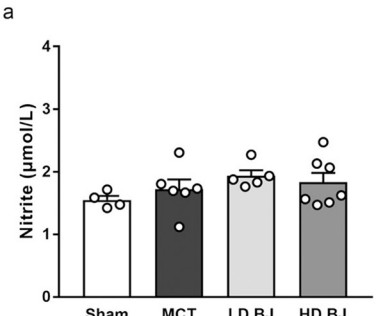 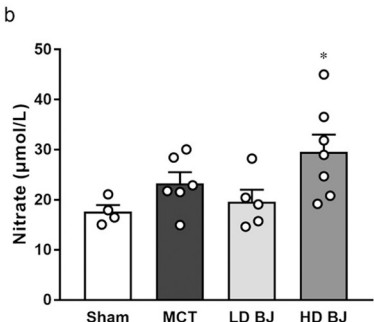 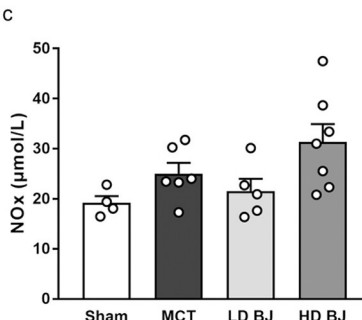

**Fig 2. Effects of nitrate-rich BJ supplementation on plasma NO levels in MCT-injected rats.** (a) nitrite. (b) nitrate. (c) NOx. Each point represents individual value; each column and bar represent the mean ± SEM values of 4–7 experiments. *p < 0.05, compared to the sham group. Statistical analysis was performed using one-way analysis of variance with Sidak post hoc test. Abbreviations: BJ, beetroot juice; HD, high-dose; LD, low-dose; MCT, monocrotaline; NOx, nitrite plus nitrate.

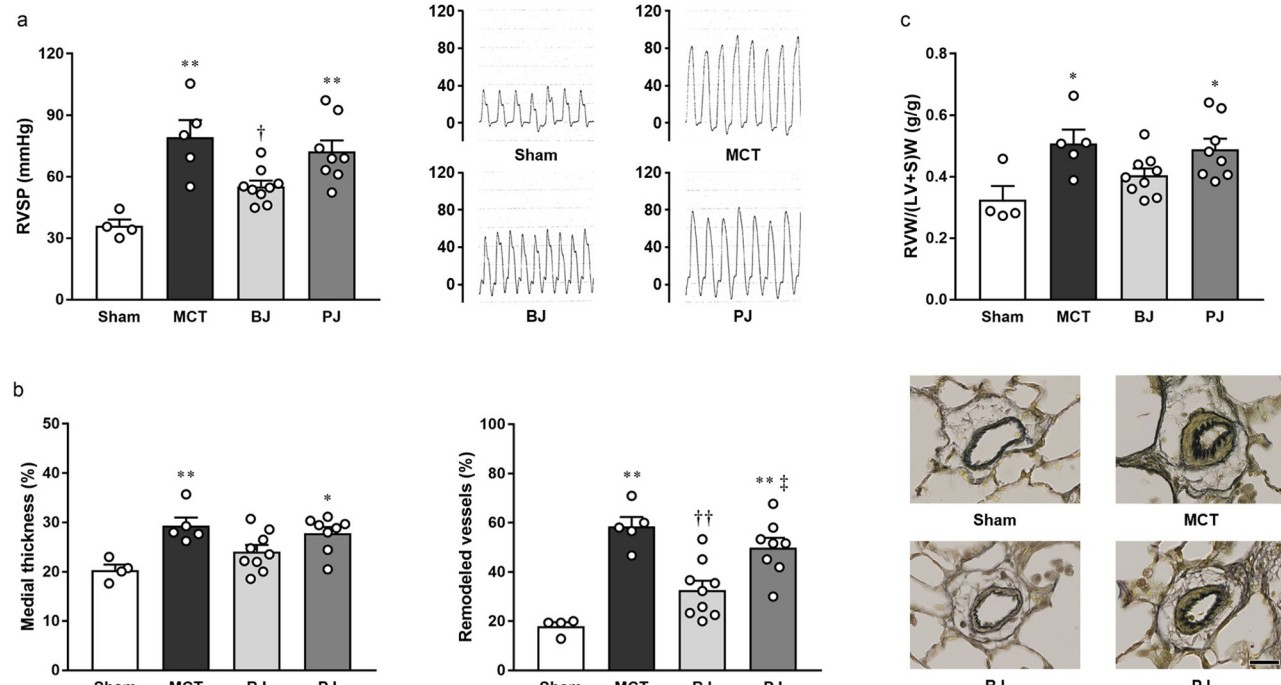

**Fig 3. Effects of nitrate-depleted BJ supplementation on PH symptoms in MCT-injected rats.** (a) RVSP (left panel) and typical tracings (right panel). (b) Pulmonary arterial medial thickening (left panel), remodeling (middle panel), and typical images (right panel; scale bar, 25 μm). (c) RVW/(LV+S)W. Each point represents individual value; each column and bar represent the mean ± SEM values of 4–9 experiments. *p < 0.05 and **p < 0.01, compared to the sham group; †p < 0.05 and ††p < 0.01, compared to the MCT group; ‡p < 0.05, compared to the BJ group. Statistical analysis was performed using one-way analysis of variance with Sidak post hoc test. Abbreviations: BJ, beetroot juice; MCT, monocrotaline; PJ, placebo juice; RVSP, right ventricular systolic pressure; RVW/(LV+S)W, right ventricle-to-left ventricle plus septum weight ratio.

## Discussion

In the aspect of health promotion, BJ has gained a lot of attention in the medical field, which has thus led to its recent adoption for use in therapy of some cardiovascular diseases such as

**Table 2. Effects of nitrate-depleted BJ supplementation on various parameters.**

|  | Sham (n = 4) | MCT (n = 5) | BJ (n = 9) | PJ (n = 8) |
|---|---|---|---|---|
| BW (g) | 389 ± 13 | 333 ± 19 | 340 ± 7 | 362 ± 12 |
| HW/BW (g/kg) | 2.58 ± 0.08 | 2.86 ± 0.15 | 2.82 ± 0.04 | 3.00 ± 0.09* |
| RVW/BW (g/kg) | 0.58 ± 0.06 | 0.89 ± 0.09 | 0.75 ± 0.04 | 0.91 ± 0.06* |
| (LV+S)W/BW (g/kg) | 1.81 ± 0.08 | 1.75 ± 0.04 | 1.87 ± 0.02 | 1.86 ± 0.03 |
| LW/BW (g/kg) | 3.50 ± 0.08 | 6.44 ± 0.91** | 5.42 ± 0.19* | 5.76 ± 0.16** |
| HR (bpm) | 357 ± 31 | 370 ± 21 | 395 ± 14 | 365 ± 10 |
| MAP (mmHg) | 87 ± 5 | 87 ± 6 | 91 ± 7 | 79 ± 5 |

Data are the mean ± SEM values of 4–9 experiments.

*p < 0.05 and

**p < 0.01, compared to the sham group.

Statistical analysis was performed using one-way analysis of variance with Sidak post hoc test. Abbreviations: MCT, monocrotaline; BJ, beetroot juice; PJ, placebo juice; BW, body weight; HW, heart weight; RVW, right ventricular weight; (LV+S)W, left ventricular plus septum weight; LW, lung weight; HR, heart rate; MAP, mean arterial pressure.

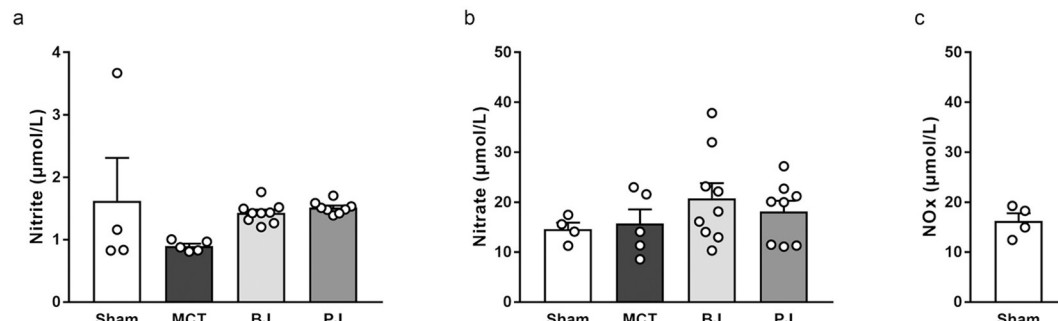

**Fig 4. Effects of nitrate-depleted BJ supplementation on plasma NO levels in MCT-injected rats.** (a) nitrite. (b) nitrate. (c) NOx. Each point represents individual value; each column and bar represent the mean ± SEM values of 4–9 experiments. Statistical analysis was performed using one-way analysis of variance with Sidak post hoc test. Abbreviations: BJ, beetroot juice; PJ, placebo juice; MCT, monocrotaline; NOx, nitrite plus nitrate.

hypertension [22, 23], heart failure [22, 24], atherosclerosis [25], coronary artery disease [26], and peripheral artery disease [27]. Notwithstanding, the available evidence is not convincing enough to substantiate the use of BJ as a therapy in the management of PH yet. The results obtained in this study, however, provide important information to curb this challenge.

Due to the variety in the available BJ products, a difference in the composition and content of nutrients is expected to exist, suggesting that efficacy might differ with respective varieties. However, following the previous study wherein BJ from a different manufacturer was used [16], this study substantiated further the finding that the efficacy observed when managing PH is dependent on the amount of BJ supplementation used. That is, MCT-induced RVSP elevation, pulmonary vascular remodeling, and right ventricular hypertrophy were alleviated in rats supplemented with BJ containing a less amount of nitrate (approximately 100 µmol/kg/day), but not in rats supplemented with BJ containing a high amount of nitrate (approximately 500 µmol/kg/day). While the plasma levels of NO metabolites in the former remained significantly unchanged, there was a marked increase in the nitrate levels of the latter. The finding that BJ supplementation prevented the development of PH without increasing plasma NO metabolites is also in line with a previous report [16].

The scientific novelty of this study lies in the identification of a causative component for BJ-induced beneficial effects on PH. Apart from the nitrate, BJ has other contents that possess health benefits [8]. This makes it imperative to identify the content(s) actually responsible for BJ-induced effects. Interestingly, recent reports indicate the importance of nitrate-independent effects of BJ [28, 29]. In this study, the beneficial effects of BJ supplementation (nitrate intake: approximately 100 µmol/kg/day) on PH symptoms, such as RVSP elevation, pulmonary vascular remodeling, and right ventricular hypertrophy, were lost following the extraction of nitrate from the juice. These findings suggest that BJ ingestion halt the progression of PH in a nitrate-dependent manner. As supporting evidence for this view, Henrohn et al. have reported that right ventricular function tends to be improved, though not significantly, in PH patients treated with nitrate-rich BJ (nitrate intake: approximately 250 µmol/kg/day) compared to that in those treated with nitrate-depleted BJ [17]. However, our study showed that neither supplementation with nitrate-rich BJ nor nitrate-depleted BJ affected the plasma levels of NO metabolites; this is of great concern.

Considering the findings obtained using nitrate-depleted BJ, the preventive effects of BJ on PH are most likely due to the nitrate, even though the effects observed were not accompanied by an increase in the plasma level of NO metabolites. Therefore, this brings up the question if there was a rise in the level of NO in any tissue other than the blood. For example, there is a

possibility that NO was increased in the pulmonary circulation. The nitrate-nitrite-NO pathway is stimulated under hypoxic conditions; this is because the expression and activity of xanthine oxidoreductase, a key enzyme that catalyzes the reduction of nitrite to NO, are enhanced in response to hypoxia [30]. Such a phenomenon has in fact been observed in the lung and failing right ventricle in a rat model of MCT-induced PH [31, 32]. Even though the levels of NO metabolites in the tissues, as mentioned earlier, were not examined in this study, the BJ-derived nitrate could have led to an increase in the level of NO within the pulmonary circulation.

Beetroot is currently being applied as a functional food. Interestingly, it has been suggested that the protective effects of functional foods on cardiovascular diseases are mediated by epigenomic mechanisms [33]. In this regard, NO functions as an epigenetic regulator of gene expression and cell phenotype [34]. As epigenetic dysregulation, including DNA hypermethylation, histone hyperacetylation, and altered microRNAs expression, play a role in the progression of PH [4], NO replenishment might improve these conditions. Whether or not BJ supplementation induces beneficial effects on PH by affecting the epigenome is a very interesting issue.

The results from other studies provide the possibility that components other than nitrate also played a small role in BJ-induced beneficial effects. For example, β-carotene, a carotenoid, has been shown to be capable to alleviate MCT-induced lung injury [35]. In addition, quercetin, a flavonoid, has been documented to improve PH symptoms in MCT-injected rats [36]. These ingredients are found at high levels in beetroot [8].

A possible reason for BJ supplementation at a high dose to mask the beneficial effects on PH has been previously discussed fully [16] where we considered the nitrate to be also responsible for this masking effect. In general, overconsumption of nitrate has rather negative effects on cardiovascular homeostasis [37, 38]. In this regard, a recent study showed that treatment of MCT-injected rats with inorganic nitrate at 300 or 1000 μmol/kg/day increased plasma nitrite and nitrate levels with no observable effect on right ventricular hypertrophy [39]. Hence, the effective dose range of nitrate on PH may be very narrow, and a dose that does not increase the systemic NO level would be preferable. This theory is further supported by our findings that with an increase in the plasma NOx level, the noted improvement in PH with the use of isosorbide mononitrate when there was no rise in NOx level, was no longer be observed [40]. However, dissimilarity exists in the pharmacokinetics of dietary nitrate between rodents and humans—the circulating nitrate does not accumulate in the saliva of rodents whereas that in humans is significantly concentrated in the saliva [41]. Therefore, optimal doses may vary between rodents and humans.

Three important pathways (the NO pathway, the endothelin pathway, and the prostacyclin pathway) have been identified in PH and combination therapy targeting these pathways is regarded as a treatment option that should be considered [42]. For example, combination therapy of endothelin receptor antagonist and phosphodiesterase-5 inhibitor provides synergistic effects in patients with PH [43]. Likewise, the combination of prostacyclin receptor agonist and phosphodiesterase-5 inhibitor results in better outcomes than monotherapy [44]. As it was demonstrated in this study that BJ ingestion suppresses the development of PH in a nitrate-dependent manner, BJ might also be effective in combination with drugs targeting the endothelin and prostacyclin pathways from a clinical perspective. That is, BJ intake would be recommended as a supplementary treatment option for PH patients receiving endothelin receptor antagonist or prostacyclin receptor agonist therapy. Additionally, the fact that BJ intake is cost effective as a therapeutic intervention could render it generally acceptable as a treatment option. Although only a single study has been done in this regard [17], the ongoing

(clinicalTrials.gov identifier: NCT01682356) and future clinical trials will definitely address the usefulness of BJ as a supplement in treating PH.

There are several limitations in this study. First, as mentioned earlier, we did not examine whether BJ supplementation increases NO level in pulmonary circulation in PH. As PH is accompanied by a decrease in NO bioavailability in pulmonary circulation [4], this issue is of great importance in further exploring the therapeutic value of BJ. Second, another major limitation is the lack of information on the mechanism by which BJ-derived nitrate exerts

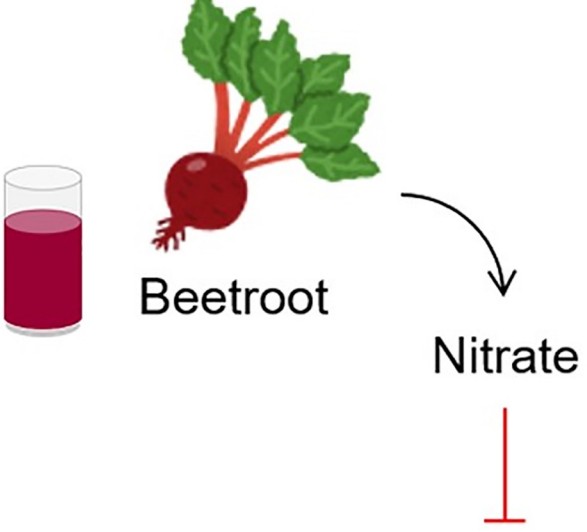

Beetroot

Nitrate

Pulmonary hypertension

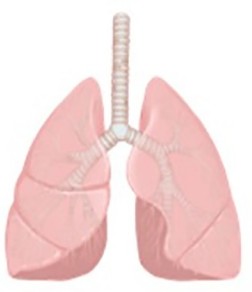 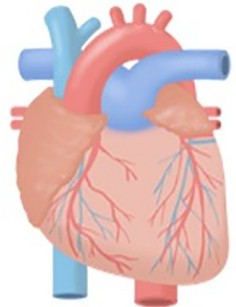

- Pulmonary arterial remodeling

- RVSP elevation
- Right ventricular hypertrophy

**Fig 5. Schematic representation of BJ-induced beneficial effects on PH.** Abbreviations: BJ, beetroot juice; PH, pulmonary hypertension; RVSP, right ventricular systolic pressure.

beneficial effects on PH. Although nitrite and NO, reduction products of nitrate, have pulmonary arterial dilating, anti-proliferative, and anti-inflammatory effects [31, 45, 46], it remains unclear whether BJ-derived nitrate improves PH symptom through one or combinations of these effects. Interestingly, it has been demonstrated that PH is affected by the intestinal environment [47, 48]. BJ ingestion and dietary nitrate have been demonstrated to modulate the gut microflora and its metabolic activity [49, 50]. Therefore, it is possible that BJ-induced beneficial effects on PH are partially via the attenuation of gut pathology. In this theory, an increase in systemic NO level is not always necessary, and this is consistent with our results. Future research should focus on how BJ-derived nitrate suppresses PH.

In summary, it was demonstrated for the first time that ingestion of a suitable amount of BJ protects against PH symptoms such as RVSP elevation, pulmonary vascular remodeling, and right ventricular hypertrophy, mostly in a nitrate-dependent manner (Fig 5). Thus, this finding provides a strong evidence to substantiate the clinical application of BJ to patients with PH.

## Supporting information

**S1 File.**
(XLSX)

## Acknowledgments

The authors thank Kana Iesaki, Yuka Murata, Shun Kusaka, and Miyu Nakajima (Laboratory of Pathological and Molecular Pharmacology, Osaka University of Pharmaceutical Sciences) for their assistance.

## Author Contributions

**Conceptualization:** Masashi Tawa, Mamoru Ohkita, Yasuo Matsumura.

**Formal analysis:** Masashi Tawa, Rikako Nagata, Yuiko Sumi.

**Investigation:** Masashi Tawa, Rikako Nagata, Yuiko Sumi, Keisuke Nakagawa, Tatsuya Sawano.

**Supervision:** Yasuo Matsumura.

**Writing – original draft:** Masashi Tawa.

**Writing – review & editing:** Masashi Tawa, Rikako Nagata, Yuiko Sumi, Keisuke Nakagawa, Tatsuya Sawano, Mamoru Ohkita, Yasuo Matsumura.

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
