## [Decision Letter · Decision Letter 0]

24 Sep 2020

PONE-D-20-22725

Preventive effects of nitrate-rich beetroot juice supplementation on monocrotaline-induced pulmonary hypertension in rats

PLOS ONE

Dear Dr. Tawa,

Thank you for submitting your manuscript to PLOS ONE. After careful consideration, we feel that it has merit but does not fully meet PLOS ONE’s publication criteria as it currently stands. Therefore, we invite you to submit a revised version of the manuscript that addresses the points raised during the review process.

The manuscript was reviewed by three referees, all of whom have recommended additonal experiments and mechanistic studies to strengthen the findings. Kindly refer to the comments below from the reviewers for more details.

We look forward to receiving your revised manuscript.

Kind regards,

Vinayak Shenoy

Academic Editor

PLOS ONE

Journal Requirements:

2. Please ensure you have thoroughly discussed any potential limitations of this study within the Discussion section.

Reviewers' comments:

Reviewer's Responses to Questions

**Comments to the Author**

1. Is the manuscript technically sound, and do the data support the conclusions?

Reviewer #1: Yes

Reviewer #2: Yes

Reviewer #3: Yes

2. Has the statistical analysis been performed appropriately and rigorously? 

Reviewer #1: Yes

Reviewer #2: No

Reviewer #3: Yes

3. Have the authors made all data underlying the findings in their manuscript fully available?

Reviewer #1: Yes

Reviewer #2: No

Reviewer #3: Yes

4. Is the manuscript presented in an intelligible fashion and written in standard English?

Reviewer #1: Yes

Reviewer #2: Yes

Reviewer #3: Yes

5. Review Comments to the Author

Reviewer #1: In this study by Masashi Tawa et al, “Preventive effects of nitrate-rich beetroot juice supplementation on monocrotaline-induced pulmonary hypertension in rats” the authors provide evidence that Beetroot juice (BJ) attenuates pulmonary hypertension in a rat model induced by MCT injection. The study is straightforward, and the data are convincing, however there are a few points that could be considered:

The manuscript would be greatly strengthened by analysis of immune/inflammatory cells in the lung, either by flow cytometry or immunostaining or PCR.

Is the proposed mechanism unique to the MCT model of PH or applicable to another etiological factor mediated PH? Have tried in another model of PH?

The authors have shown a preventative effect of BJ on development of PH. Translationally, how early would this need to be instituted to prevent or treat PH? Can PH be revered with BJ?

It would be better to have a Figure with raw tracings of RVSP and the parameters (All together in one frame)

Since right ventricle mainly affected in PH, any data to show the right ventricle pathology/collagen staining with picrosirius?

BJ was given with water, suggest increase some discussion on BJ’s impact on gut or its microbiota if there is any possible axis in PH.

Suggest increased discussion of other possible indirect mechanisms, and any data refuting/suggesting them

Was the lung inflated during the perfusion before the staining?

A graphical abstract would be a great idea, showing the mechanisms of beneficial effects BJ?

Reviewer #2: The manuscript by Tawa et al. shows that the dietary intake of beetroot juice reduced monocrotaline-induced increase in right ventricular systolic pressure, pulmonary arterial medial thickening, and right ventricular hypertrophy. The authors further, demonstrated that this effect of beetroot juice was mediated by low dose nitrate (1.3 mmol/L). However, higher dose of nitrate (4.3 mmol/L) did not alleviate monocrotaline-induced pulmonary dysfunction.

The manuscript is very well written. However, the data presented in the manuscript is insufficient to comprehensively prove the hypothesis.

1. Firstly, the study is not novel as a similar study has been done in humans "Effects of Oral Supplementation With Nitrate-Rich Beetroot Juice in Patients With Pulmonary Arterial Hypertension—Results From BEET-PAH, an Exploratory Randomized, Double-Blind, Placebo-Controlled, Crossover Study" by Henrohn et al. Oct 2018. Therefore, I would recommend the authors to expand on the current findings to explore additional parameters of endothelial function that may be impacted by beetroot supplementation.

2. The authors also need to measure readouts of nitric oxide such as nitrate/nitrite levels post monocrotaline and beetroot administration in animals.

3. Histology images of pulmonary artery should be added to the manuscript.

4. The experiments should have at least 5 replicates.

Reviewer #3: There is now a substantial body of data suggesting nitrate supplementation can have a beneficial effect on heamodynamics. Tawa et al have previously shown that beetroot juice supplementation can ameliorate PH in monocrotaline induced PH in rats. In this study they attempt to add to previous data by demonstrating the beneficial effects of beetroot juice are dependent on its nitrate content. To this effect, they demonstrate nitrate depleted beetroot juice is ineffective in MCT induced PH.

Major comments.

The first experiment replicates previously published data with two doses of Beetroot juice. As previously published low dose beetroot juice is more effective in preventing PH than highdose beetroot juice. However, this interesting finding is not followed up.

The second experiment focuses on nitrate depletion, and nitrate depleted beetroot juice is ineffective in reducing PH. However no nitrate/nitrite plasma or urine levels are shown. Especially as there appears to be a dose dependant reaction to nitrate, it would be pertinent to the experiment to show systemic nitrate/nitrite levels in beetroot supplemented, depleted beetroot and naive animals.

Pulmonary vascular remodelling is determined solely by measuring the medial thickness in chosen resistance pulmonary arteries. However the percentage of muscularised vessels is known to be increased in PH, and it would be helpful to this parameter (non, partial, fully muscularised percentages) when discussing effects on vascular remodelling.

The paper is also lacking in mechanism. Does nitrate depletion lead to proliferation of SMC, or reduced inflammation in vessel walls or it purely a dilation effect of increased NO availability? Please add in some mechanistic studies.

6. PLOS authors have the option to publish the peer review history of their article (what does this mean?). If published, this will include your full peer review and any attached files.

Reviewer #1: **Yes: **Ravindra K Sharma

Reviewer #2: No

Reviewer #3: No

---

## [Author Response · Author response to Decision Letter 0]

18 Dec 2020

Responses to Reviewer #1

We are truly grateful for your careful reading of our manuscript and for your valuable comments. Our changes in the revised manuscript have been denoted in red. We hope our responses have sufficiently clarified the reviewers’ concerns.

The manuscript would be greatly strengthened by analysis of immune/inflammatory cells in the lung, either by flow cytometry or immunostaining or PCR.

As you mentioned, this study did not address how BJ-derived nitrate prevents PH symptoms. Since this is a critical limitation of this study, a related description has been added in the Discussion (see line 308–317).

Is the proposed mechanism unique to the MCT model of PH or applicable to another etiological factor mediated PH? Have tried in another model of PH?

We have never tried in another model of PH and there is also no report examining the effects of BJ supplementation on PH models other than MCT. Since dietary inorganic nitrate has been reported to be effective for hypoxia- and bleomycin-induced PH (Circulation. 2012; 125: 2922–2932), BJ supplementation might similarly have protective effects against PH models other than MCT. As you mentioned, this truth must be clarified in the future.

The authors have shown a preventative effect of BJ on development of PH. Translationally, how early would this need to be instituted to prevent or treat PH? Can PH be revered with BJ?

Your comment points out a very important issue. Unfortunately, from our findings so far, we can only say that the intervention should start as soon as possible after the onset of PH. In addition, we reported that BJ supplementation is ineffective when started after PH had developed to some extent; however, only one dose was examined in that study (Am J Hypertens. 2019; 32: 216–222). That is, there is a possibility that a suitable amount of BJ (or nitrate) to reverse PH is different from that to prevent PH. Whether or not BJ supplementation can reverse PH is still under discussion.

It would be better to have a Figure with raw tracings of RVSP and the parameters (All together in one frame).

According to your comment, typical tracings of RVSP have been added in the Figure (see Figs 1a and 3a).

Since right ventricle mainly affected in PH, any data to show the right ventricle pathology/collagen staining with picrosirius?

In this study, we did not make paraffin blocks of right ventricle for histological analysis. Therefore, we are sorry but do not have any data to show morphological alterations of right ventricle.

BJ was given with water, suggest increase some discussion on BJ’s impact on gut or its microbiota if there is any possible axis in PH.

As you mentioned, it cannot be denied that BJ supplementation may have been effective for PH by improving the intestinal environment. Therefore, according to your comment, a related description has been added in the Discussion (see line 312–316).

Suggest increased discussion of other possible indirect mechanisms, and any data refuting/suggesting them.

According to your comment, a related description has been added in the Discussion (see line 274–278).

Was the lung inflated during the perfusion before the staining?

Yes. We are sorry for our insufficient description and the corresponding part has been revised so that the readers can understand the exact procedure (see line 118).

A graphical abstract would be a great idea, showing the mechanisms of beneficial effects BJ?

We agree your comment. However, this journal does not give an option to provide a graphical abstract. Therefore, we have added a summary scheme as the figure in the text (see Fig 5 and line 320).

 

Responses to Reviewer #2

We are truly grateful for your careful reading of our manuscript and for your valuable comments. Our changes in the revised manuscript have been denoted in red. We hope our responses have sufficiently clarified the reviewers’ concerns.

1. Firstly, the study is not novel as a similar study has been done in humans “Effects of Oral Supplementation With Nitrate-Rich Beetroot Juice in Patients With Pulmonary Arterial Hypertension−Results From BEET-PAH, an Exploratory Randomized, Double-Blind, Placebo-Controlled, Crossover Study” by Henrohn et al. Oct 2018. Therefore, I would recommend the authors to expand on the current findings to explore additional parameters of endothelial function that may be impacted by beetroot supplementation.

This study provides new evidence on the nitrate-dependent inhibitory effect of BJ on PH-induced pulmonary arterial remodeling. Although pulmonary arterial remodeling is an important index of PH, the report by Henrohn et al. did not address this issue. Therefore, we believe this study is of sufficient scientific novelty. As you mentioned, the influence of BJ supplementation on pulmonary endothelial function is very intriguing. Since we are familiar with experiments on evaluation of pulmonary vascular function (Am J Hypertens. 2020; 33: 305–309, Hypertens Res. 2020; 43: 178–185, J Pharmacol Sci. 2019; 140: 43–47, Life Sci. 2018; 203: 203–209), we are planning to perform such experiments. However, it takes a lot of time and animals to carry out the experiments, hence we would like to set those as the next theme. Thank you for your understanding.

2. The authors also need to measure readouts of nitric oxide such as nitrate/nitrite levels post monocrotaline and beetroot administration in animals.

According to your comment, data on plasma nitrite, nitrate, and their sum (NOx) levels and related descriptions have been added (see Figs 2 and 4, lines 32–34, 38–40, 102–103, 135–143, 179–191, 220–228, 245–248, 260–273, and 285–286). Although increases in those levels were not observed even by nitrate-rich BJ supplementation, we consider the fact that nitrate-depleted BJ was ineffective is the best evidence showing that BJ exerts beneficial effects in a nitrate-dependent manner.

3. Histology images of pulmonary artery should be added to the manuscript.

According to your comment, typical images of pulmonary artery have been added in the Figure (see Figs 1b and 3b).

4. The experiments should have at least 5 replicates.

In this study, it is the sham group that the sample size corresponds to less than 5. However, the sum of the first and second experiments gives 8 samples. In addition, we have collected a lot of sham data so far, and the sham data has little variation (Am J Hypertens. 2019; 32: 216–222, Life Sci. 2018; 203: 203–209). Based on these lines, we have decided that the sample sizes in the sham groups were acceptable limits. We believe our idea follows the 3R principles “replace/reduce/refine” for animal use for scientific purposes, and thank you for your understanding.

 

Responses to Reviewer #3

We are truly grateful for your careful reading of our manuscript and for your valuable comments. Our changes in the revised manuscript have been denoted in red. We hope our responses have sufficiently clarified the reviewers’ concerns.

The first experiment replicates previously published data with two doses of Beetroot juice. As previously published low dose beetroot juice is more effective in preventing PH than high dose beetroot juice. However, this interesting finding is not followed up.

As you mentioned, it is an important issue why high-dose BJ supplementation was ineffective for MCT-induced PH. As described in the Discussion (see line 279–292), we think the reason for this lies in nitrate in BJ. However, this is difficult to prove because it cannot be demonstrated with nitrate-depleted BJ; unless PH is worsened by nitrate-rich BJ supplementation, it is of little significance to compare with its placebo. We do not know if it becomes a supplement to support the above hypothesis, but data on plasma nitrite, nitrate, and their sum (NOx) levels have been added (see Figs 2 and 4, lines 32–34, 38–40, 102–103, 135–143, 179–191, 220–228, 245–248, 260–273, and 285–286).

The second experiment focuses on nitrate depletion, and nitrate depleted beetroot juice is ineffective in reducing PH. However, no nitrate/nitrite plasma or urine levels are shown. Especially as there appears to be a dose dependent reaction to nitrate, it would be pertinent to the experiment to show systemic nitrate/nitrite levels in beetroot supplemented, depleted beetroot and naïve animals.

According to your comment, data on plasma nitrite, nitrate, and NOx levels and related descriptions have been added (see Figs 2 and 4, lines 32–34, 38–40, 102–103, 135–143, 179–191, 220–228, 245–248, 260–273, and 285–286). Although increases in those levels were not observed even by nitrate-rich BJ supplementation, we consider the fact that nitrate-depleted BJ was ineffective is the best evidence showing that BJ exerts beneficial effects in a nitrate-dependent manner.

Pulmonary vascular remodelling is determined solely by measuring the medial thickness in chosen resistance pulmonary arteries. However the percentage of muscularised vessels is known to be increased in PH, and it would be helpful to this parameter (non, partial, fully muscularised percentages) when discussing effects on vascular remodelling.

According to your comment, data on the percentage of remodeled vessels and related descriptions have been added (see Figs 1b and 3b, lines 30–31, 128–133, 154–155, and 196–197).

The paper is also lacking in mechanism. Does nitrate depletion lead to proliferation of SMC, or reduced inflammation in vessel walls or it purely a dilation effect of increased NO availability? Please add in some mechanistic studies.

The results in this study show that nitrate is involved in the underlying mechanism by which BJ supplementation prevents PH development. Therefore, we believe this study has an aspect of the mechanistic study. However, as you mentioned, this study did not address how BJ-derived nitrate prevents PH symptoms. Since this is a critical limitation of this study, a related description has been added in the Discussion (see line 308–317).

---

## [Decision Letter · Decision Letter 1]

10 Mar 2021

PONE-D-20-22725R1

Preventive effects of nitrate-rich beetroot juice supplementation on monocrotaline-induced pulmonary hypertension in rats

PLOS ONE

Dear Dr. Tawa,

Thank you for submitting your manuscript to PLOS ONE. After careful consideration, we feel that it has merit but does not fully meet PLOS ONE’s publication criteria as it currently stands. Therefore, we invite you to submit a revised version of the manuscript that addresses the points raised during the review process.

ACADEMIC EDITOR: All issues raised by editor and reviewer are required to clearly support the conclusions.

We look forward to receiving your revised manuscript.

Kind regards,

Vincenzo Lionetti, M.D., PhD

Academic Editor

PLOS ONE

Additional Editor Comments (if provided):

In light of revised version of the article, I would highlight the following major issues:

1) It is important to evaluate the activity and protein expression of endothelial nitric oxide synthase (NOS) in lung tissue and pulmonary vessels in order to properly support the conclusions. Indeed, we don't have still evidence of pulmonary modulation of "good" NO production compared to "bad" NO production that depends on inducible NOS.

2) The authors should better characterize cardiac phenotype by measuring cardiomyocyte size and myocardial collagen deposits.

3) Previous study highlighted the importance of functional food compounds in cardioprotection through action on the epigenome (Eur Heart J. 2019 Feb 14;40(7):575-582). Since NO exerts epigenetic modifications, discussion on epigenetic regulation of expression of adaptive genes by NO should be added.

Reviewers' comments:

Reviewer's Responses to Questions

**Comments to the Author**

1. If the authors have adequately addressed your comments raised in a previous round of review and you feel that this manuscript is now acceptable for publication, you may indicate that here to bypass the “Comments to the Author” section, enter your conflict of interest statement in the “Confidential to Editor” section, and submit your "Accept" recommendation.

Reviewer #2: All comments have been addressed

Reviewer #3: (No Response)

Reviewer #4: All comments have been addressed

2. Is the manuscript technically sound, and do the data support the conclusions?

Reviewer #2: Yes

Reviewer #3: Yes

Reviewer #4: Yes

3. Has the statistical analysis been performed appropriately and rigorously? 

Reviewer #2: Yes

Reviewer #3: Yes

Reviewer #4: Yes

4. Have the authors made all data underlying the findings in their manuscript fully available?

Reviewer #2: Yes

Reviewer #3: Yes

Reviewer #4: Yes

5. Is the manuscript presented in an intelligible fashion and written in standard English?

Reviewer #2: Yes

Reviewer #3: Yes

Reviewer #4: Yes

6. Review Comments to the Author

Reviewer #2: (No Response)

Reviewer #3: The authors have made an attempt to answer the questions raised in the first review by performing two additional analysis but no new experiments.

Firstly they have added pulmonary vascular remodelling data as requested. This shows that LD BJ and nitrate rich BJ both reduce the number of muscularised small pulmonary vessels while HD BJ and nitrate depleted BJ do not. This provides addition evidence that nitrate derived from BJ is efficacious in reducing PH induced remodelling (their previous data looking only at medial thickness showed no significant reduction).

Secondly they have added NOx plasma measurements as requested by multiple reviewers. The data show no significant changes in any NOx parameters in the LD BJ and the nitrate rich BJ supplementation groups. The only increases in plasma NOx is seen in the HD BJ group which paradoxically does not show any amelioration of PH. While it is reassuring that the experimental nitrate depleted control group shows no change from MCT alone, there is still insufficient evidence for mechanism. The authors suggest that it is local concentrations of nitrate rather than systemic that could be the driving factor for physiological effect but show no data to that effect. I feel adding lung NOx (and cGMP) concentrations in tissue would would add some mechanism to this paper.

As noted the paper still lacks novel mechanistic insight. Previous studies by various groups referenced in this paper have suggested BJ /nitrate supplementation is effective though the NO-cGMP pathway affecting proliferation/dilation/endothelial function/inflammation etc. However there is no data in this paper to suggest that is what they are seeing. The authors have also suggested in the discussion that there could be effects on gut flora, but there is no attempt to include any data on that theory. The new schematic (Figure 5) is very general, and it would be better to add more specific pathways.

Reviewer #4: In this study by Masashi Tawa et al, “Preventive effects of nitrate-rich beetroot juice supplementation on monocrotaline-induced pulmonary hypertension in rats” the authors provide evidence that Beetroot juice (BJ) attenuates pulmonary hypertension in a rat model. This is revise manuscript, and the authors have addressed all the comments.

7. PLOS authors have the option to publish the peer review history of their article (what does this mean?). If published, this will include your full peer review and any attached files.

Reviewer #2: No

Reviewer #3: No

Reviewer #4: **Yes: **Ravindra K Sharma

---

## [Author Response · Author response to Decision Letter 1]

16 Mar 2021

Responses to Editor

We are truly grateful for your careful reading of our manuscript and for your valuable comments. Our changes in the revised manuscript have been denoted in purple. We hope our responses have sufficiently clarified the editor’s concerns.

1) It is important to evaluate the activity and protein expression of endothelial nitric oxide synthase (NOS) in lung tissue and pulmonary vessels in order to properly support the conclusions. Indeed, we don't have still evidence of pulmonary modulation of "good" NO production compared to "bad" NO production that depends on inducible NOS.

Inorganic nitrate present in BJ produces NO in an eNOS-independent manner; therefore, we did not focus on the eNOS/NO pathway and did not measure eNOS activity and expression in this study. As you mentioned, excessive NO is a mediator of cytotoxicity in pulmonary circulation, but it is well known that PH is accompanied by a decrease in eNOS-derived NO bioavailability. Indeed, many studies have found that pulmonary eNOS activation by eNOS gene therapy (Am J Respir Cell Mol Biol. 2006; 35: 182–189, FASEB J. 2006; 20: 2594–2596) and chronic administration of L-arginine (Am J Physiol Endocrinol Metab. 2010; 298: E1131–E1139, Eur J Pharm Sci. 2014; 62: 161–170) and tetrahydrobiopterin (Cardiovasc Ther. 2018; 36: e12312) ameliorates the symptom of PH. Therefore, it is assumed that “good” NO production, regardless of endogenous generation or exogenous administration, is beneficial in the management of PH.

2) The authors should better characterize cardiac phenotype by measuring cardiomyocyte size and myocardial collagen deposits.

As mentioned in the response to the previous reviewers’ comments, we did not make paraffin blocks of right ventricle for histological analysis. Therefore, we are sorry but do not have data on cardiomyocyte size and myocardial collagen deposits.

3) Previous study highlighted the importance of functional food compounds in cardioprotection through action on the epigenome (Eur Heart J. 2019 Feb 14;40(7):575-582). Since NO exerts epigenetic modifications, discussion on epigenetic regulation of expression of adaptive genes by NO should be added.

According to your comment, a related description has been added in the Discussion (see lines 274–281).

 

Responses to Reviewer #3

We are truly grateful for your careful reading of our manuscript and for your valuable comments. Our changes in the revised manuscript have been denoted in purple. We hope our responses have sufficiently clarified the reviewer’s concerns.

Secondly they have added NOx plasma measurements as requested by multiple reviewers. The data show no significant changes in any NOx parameters in the LD BJ and the nitrate rich BJ supplementation groups. The only increases in plasma NOx is seen in the HD BJ group which paradoxically does not show any amelioration of PH. While it is reassuring that the experimental nitrate depleted control group shows no change from MCT alone, there is still insufficient evidence for mechanism. The authors suggest that it is local concentrations of nitrate rather than systemic that could be the driving factor for physiological effect but show no data to that effect. I feel adding lung NOx (and cGMP) concentrations in tissue would would add some mechanism to this paper.

As you mentioned, we have not been able to address the issue of whether BJ supplementation caused a local increase in NOx levels. Although we agree with your comment that the data on lung NOx levels could provide some information, we did not keep lung tissues for NOx measurements. As the lack of tissue data is a critical limitation, this limitation has been further emphasized (see lines 316–319). However, even without this data, we think that our conclusion will be well supported by the results of experiments using nitrate-depleted PJ.

As noted the paper still lacks novel mechanistic insight. Previous studies by various groups referenced in this paper have suggested BJ /nitrate supplementation is effective though the NO-cGMP pathway affecting proliferation/dilation/endothelial function/inflammation etc. However there is no data in this paper to suggest that is what they are seeing.

As described in the Introduction (see lines 46–54) and Discussion (see lines 321–322), there are already many reports showing the mechanism responsible for the protective effects of NO (or nitrate) on PH. Therefore, we did not focus on this in our study; the goal of this study was to determine a causative component for BJ-induced protective effects on PH (see lines 249–250). We think that the current data is sufficient to make a conclusion on this. In addition, we believe that our manuscript meets the publication criteria of PLOS ONE that “Authors may discuss possible implications for their results as long as these are clearly identified as hypotheses instead of conclusions”.

The authors have also suggested in the discussion that there could be effects on gut flora, but there is no attempt to include any data on that theory.

A description regarding the gut flora was added based on the reviewer’s comment, and we do not have any data on the gut flora because we did not sample and store intestines and feces. Although the statement mentions that BJ supplementation might affect the intestinal environment, it does not imply that such a conclusion has been reached. Therefore, we do not think that our discussion is an overstatement.

The new schematic (Figure 5) is very general, and it would be better to add more specific pathways.

What this study reveals is that BJ supplementation prevents the development of PH, including RVSP elevation, pulmonary arterial remodeling, and right ventricular hypertrophy, in a nitrate-dependent manner. As described in the Discussion (see lines 330–332), this finding is unprecedented and is not a generally-accepted one. In addition, this paper is not a review article; therefore, we have made a schema that summarizes only what can be said from this paper. As you mentioned, addition of more detailed information about specific pathways will be helpful, but we wonder if such a schema might have an exaggerated content. Thank you for your understanding.

---

## [Decision Letter · Decision Letter 2]

26 Mar 2021

Preventive effects of nitrate-rich beetroot juice supplementation on monocrotaline-induced pulmonary hypertension in rats

PONE-D-20-22725R2

Dear Dr. Tawa,

We’re pleased to inform you that your manuscript has been judged scientifically suitable for publication and will be formally accepted for publication once it meets all outstanding technical requirements.

Kind regards,

Vincenzo Lionetti, M.D., PhD

Academic Editor

PLOS ONE

Additional Editor Comments (optional):

Reviewers' comments:

Reviewer's Responses to Questions

**Comments to the Author**

1. If the authors have adequately addressed your comments raised in a previous round of review and you feel that this manuscript is now acceptable for publication, you may indicate that here to bypass the “Comments to the Author” section, enter your conflict of interest statement in the “Confidential to Editor” section, and submit your "Accept" recommendation.

Reviewer #2: All comments have been addressed

Reviewer #4: All comments have been addressed

2. Is the manuscript technically sound, and do the data support the conclusions?

Reviewer #2: Yes

Reviewer #4: Yes

3. Has the statistical analysis been performed appropriately and rigorously? 

Reviewer #2: Yes

Reviewer #4: Yes

4. Have the authors made all data underlying the findings in their manuscript fully available?

Reviewer #2: Yes

Reviewer #4: Yes

5. Is the manuscript presented in an intelligible fashion and written in standard English?

Reviewer #2: Yes

Reviewer #4: Yes

6. Review Comments to the Author

Reviewer #2: The authors have adequately addressed my comments and I do not have any concerns about dual publication, research ethics, or publication ethics.

Reviewer #4: The article by Masashi Tawa et al, “Preventive effects of nitrate-rich beetroot juice supplementation on monocrotaline-induced pulmonary hypertension in rats” the authors provide evidence that Beetroot juice (BJ) attenuates pulmonary hypertension in a rat model. This is revise manuscript, and the authors have addressed all the comments. The manuscript has improved significantly with this revision.

Thank you!

7. PLOS authors have the option to publish the peer review history of their article (what does this mean?). If published, this will include your full peer review and any attached files.

Reviewer #2: No

Reviewer #4: **Yes: **Ravindra K Sharma

---

## [Editor Report · Acceptance letter]

30 Mar 2021

PONE-D-20-22725R2 

Preventive effects of nitrate-rich beetroot juice supplementation on monocrotaline-induced pulmonary hypertension in rats 

Dear Dr. Tawa:

I'm pleased to inform you that your manuscript has been deemed suitable for publication in PLOS ONE. Congratulations! Your manuscript is now with our production department. 

Kind regards, 

on behalf of

Prof. Vincenzo Lionetti 

Academic Editor

PLOS ONE